# Local and thalamic origins of correlated ongoing and sensory-evoked cortical activities

Katayun Cohen-Kashi Malina[1,*,†], Boaz Mohar[1,*,†], Akiva N. Rappaport[1] & Ilan Lampl[1]

Thalamic inputs of cells in sensory cortices are outnumbered by local connections. Thus, it was suggested that robust sensory response in layer 4 emerges due to synchronized thalamic activity. To investigate the role of both inputs in the generation of correlated cortical activities, we isolated the thalamic excitatory inputs of cortical cells by optogenetically silencing cortical firing. In anaesthetized mice, we measured the correlation between isolated thalamic synaptic inputs of simultaneously patched nearby layer 4 cells of the barrel cortex. Here we report that in contrast to correlated activity of excitatory synaptic inputs in the intact cortex, isolated thalamic inputs exhibit lower variability and asynchronous spontaneous and sensory-evoked inputs. These results are further supported in awake mice when we recorded the excitatory inputs of individual cortical cells simultaneously with the local field potential in a nearby site. Our results therefore indicate that cortical synchronization emerges by intracortical coupling.

[1] Department of Neurobiology, Weizmann Institute of Science, Rehovot 76100, Israel. * These authors contributed equally to this work. † Present addresses: Department of Veterinary Resources, Weizmann institute of science, Rehovot, Israel (K.C.-K.M.); Janelia Research Campus, Howard Hughes Medical Institute, Ashburn, Virginia 20147, USA (B.M.). Correspondence and requests for materials should be addressed to I.L. (email: ilan.lampl@weizmann.ac.il).

The response of cortical cells to repeated stimuli is highly variable from trial to trial, and it is often correlated among nearby cells[1–7]. Trial-to-trial correlation of sensory responses, also known as noise correlation, can promote the saliency of neuronal responses[8–10]. However, it may reduce the capacity to carry information[11–14]. Thus, a cortical mechanism that actively decorrelates synaptic inputs could improve coding[14,15]. Since spiking mechanisms of cortical cells are thought to be highly reliable[16,17], noise correlations in spiking are likely to reflect correlated membrane potential fluctuations. Indeed, ongoing and sensory-evoked synaptic activities in nearby cortical cells are correlated both in time and magnitude[7,6,18–20]. In primary sensory cortices, layer 4 (L4) cells receive the majority of their synaptic inputs from neighbouring cortical cells[21,22]. However, they are also strongly driven by feedforward thalamic inputs[23,24]. Therefore, correlated activities between cells in L4 could either arise from common cortical noise or inherited directly from shared thalamic inputs.

In support of the first view, several studies reported that trial-to-trial variability of sensory-evoked cortical response strongly depends on the instantaneous state of cortical activity at the time of stimulation[5,9,25–28]. It was also shown that both ongoing and evoked activities can be modulated by the animal's behaviour and neuromodulators[18,19,29–31]. State-dependent modulation of noise correlation was revealed both when using paired intracellular recordings[18], or when the membrane potential was simultaneously recorded with nearby local field potential (LFP)[32,33]. Furthermore, the cortex shows slow ongoing oscillations in membrane potential after isolation from adjacent tissue[34] and when thalamus is pharmacologically inactivated[35]. Another study showed that a large component of the covarying response in the thalamus and cortex of the somatosensory system is independent of stimulus properties[36]. Taken together, these studies strongly suggest that noise correlation results from variation in cortical activity.

Alternatively, L4 variability could be inherited directly from thalamic inputs. In line with this view, it was shown that silencing cortical firing had a negligible effect on the variability of membrane potential response of L4 cells to repeated visual stimuli[4]. In addition, Bruno and Sakmann[37] proposed that the convergence of inputs from a large number of synchronous thalamic cells strongly drive L4 cells, obviating the need for cortical mechanisms such as recurrent cortical amplification to explain noise correlations.

In this study, we optogenetically silenced the cortex[38–40] while simultaneously performing whole cell and LFP recordings in awake mice and dual intracellular recordings in anaesthetized mice. This enabled us to study the contribution of thalamic and cortical excitatory synaptic inputs to the subthreshold -correlated ongoing and sensory-evoked activities in the barrel cortex. Our experiments show that cortical synchrony is not inherited from thalamic inputs but rather depends on recurrent cortical activity.

## Results

**Barrel cortex amplifies thalamic inputs.** Layer 4 (L4) cells in sensory cortices are strongly driven by feedforward thalamic inputs[23,24]. Yet the role of these inputs in the generation of correlated cortical activity was never directly tested. To determine the contribution of thalamic inputs to subthreshold correlation between L4 cells, we used an optogenetic approach to silence cortical firing while recording isolated thalamic synaptic inputs of cortical cells. To this end, we used *Gad/PV Cre* transgenic mice crossed with a *ChR2* reporter strain (Fig. 1a and 'Methods' section). Surface illumination of the somatosensory cortex (S1) with a blue LED (470 nm, ∼7 mW, LED-ON condition) activated

GABAergic cells (Fig. 1b, example cell and population data below, $0.6 \pm 0.4$ versus $80 \pm 33$ hz for LED OFF and LED ON, respectively, $n = 7$ cells, $z = -2.3664$, $P = 0.015$, Wilcoxon signed-rank test). This, in turn, inhibited the local circuitry and almost completely blocked whisker-evoked firing of excitatory cells (that is, non-ChR2 expressing cells) at all depths. The cells in the upper layers were silenced by almost 100% (example cell in the upper panel of Fig. 1c is shown during repetitive whisker stimulation; L4 population data below $0.4 \pm 0.1$ versus $0.002 \pm 0.002$ spikes per stimulus LED OFF and LED ON; $n = 7$ cells, $z = -2.3664$, $P = 0.015$ Wilcoxon signed-rank test) and the cells in the deeper layers (L5 and L6, below 500 µm,) by 92% (Fig. 1c $0.84 \pm 0.5$ versus $0.04 \pm 0.06$ spikes per stimulus LED OFF and LED ON; $n = 6$ cells, $z = -1.862$, $P = 0.0313$ Wilcoxon signed-rank test). During cortical silencing (LED ON), the recorded excitatory currents (as measured when clamping the cells at the reversal potential of inhibition, measured for each cell from the response to light) reflected remote thalamic inputs to L4 and L5 cells[38–41]. The latter also receives remote inputs from higher cortical areas[41–45]. This allowed us to estimate the amplification of these inputs by recurrent cortical circuits. For each cell, we averaged the whisker-evoked excitatory postsynaptic current (EPSC), both during intact cortical activity and when the cortex was silenced (Fig. 1d). Thalamic contribution in response to principal whisker (PW) stimulation varied considerably in individually recorded L4 and L5 cells (Fig. 1d,e). The mean relative thalamic contribution was larger in L4 than in L5 cells ($0.46 \pm 0.06$ and $0.19 \pm 0.07$, respectively, $P = 0.0198$ Mann–Whitney test, see the depth profile in Fig. 1e and the right traces in Fig. 1d), probably reflecting the greater innervation of L4 compared with L5 by thalamic fibres[41]. Importantly, the relative contribution of thalamic inputs were indistinguishable when we stimulated either the PW or the adjacent whisker (AW, Fig. 1f $P = 0.55$, $n = 18$, Wilcoxon signed-rank test). Hence, thalamic contribution to total response is unrelated to the optimality of the stimulus, similar to findings in the visual and auditory cortices[39,40].

We next verified that our manipulation allowed us to correctly isolate thalamic inputs. We first ruled out the possibility that the reduction in synaptic response was caused by shunting inhibition. Indeed, in contrast to a shunting effect, in some cells a prominent reduction in the response was recorded while no change in input resistance was measured (Fig. 1g, left example), whereas in others the response was unaffected although input resistance was reduced (Fig. 1g, right example). No significant correlation was found between the thalamic fraction and the measured change in input resistance (Fig. 1h, population data. A small trend exists, but it cannot explain the reduction of the response by shunting, as it shows minimal attenuation for cells in which input resistance was clearly reduced). In addition, by recording thalamic single units, we also ruled out the possibility that our manipulation altered the firing of ventral posteromedial nucleus cells due to cortico-thalamic feedback connections (Fig. 1h). Finally, illumination of the cortex 100 ms before whisker stimulation had no effect on the evoked currents (Supplementary Fig. 1), excluding the possibility that the reduction in EPSC is due to slow extrasynaptic activation of GABA(B) receptors[46,47]. These results indicate that the isolated thalamic synaptic inputs were not affected by cortical inactivation.

To study the contribution of thalamic inputs to the correlations between individual cells in thalamic recipient cortical layers, we performed simultaneous *in vivo* whole-cell recordings from pairs of nearby excitatory neurons in L4 (Euclidean distance < 200 µm; Fig. 2a) of anaesthetized mice (some of the cells presented in Fig. 1e were recorded as pairs). We analysed only pairs of cells for which both cells received direct thalamic inputs, as evident from

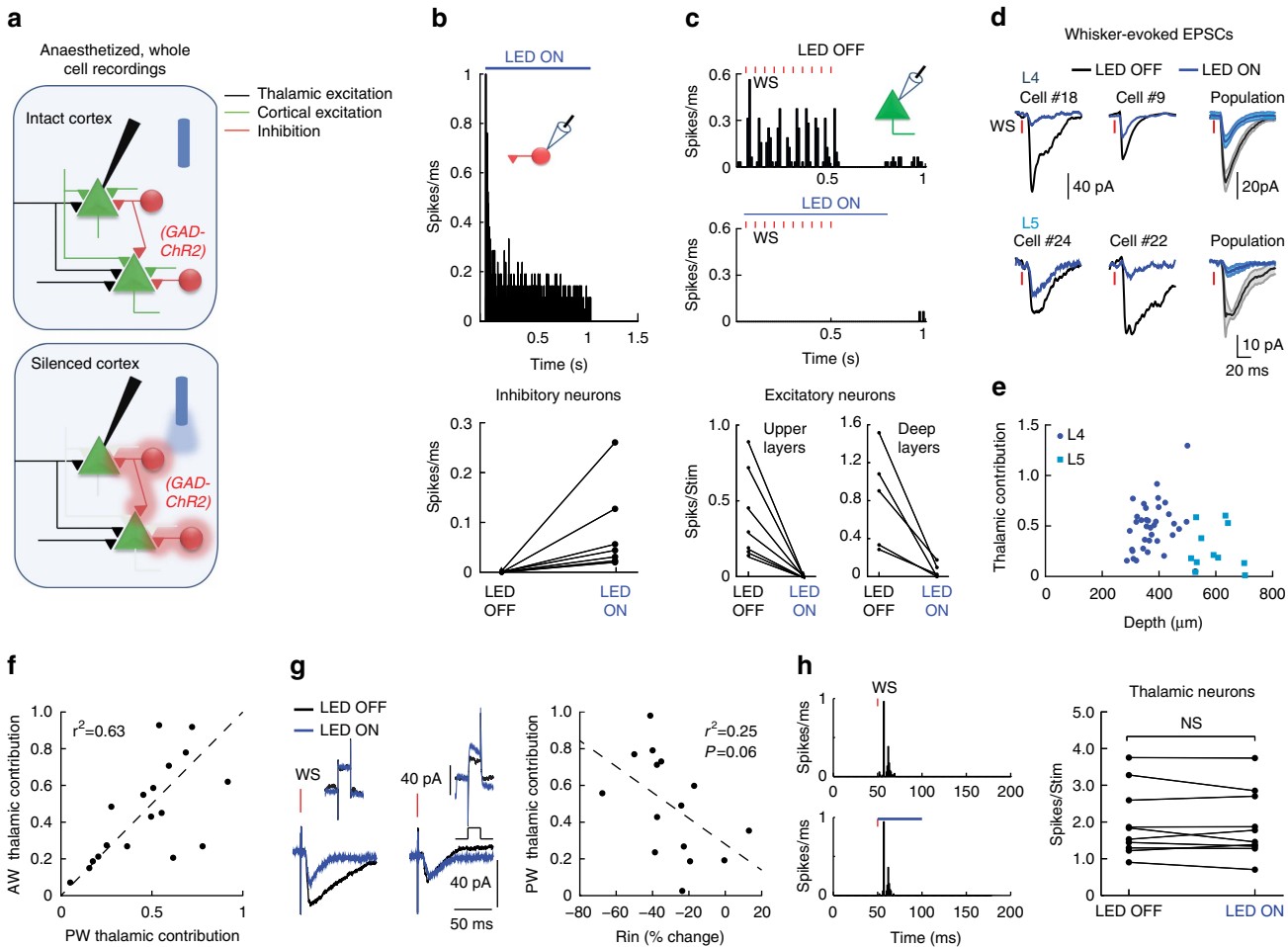

**Figure 1 | Optogenetic isolation of thalamic excitatory inputs to cortical neurons. (a)** Schematic illustration of cortical silencing experiments in anaesthetized mice. Light activates ChR2 expressing inhibitory cells, which in turn silence the firing of cortical cells. **(b)** Peri-stimulus spike time histogram (PSTH) for an example *GAD*[+]-*ChR2* cell in response to 1 s LED illumination (top) and population mean firing rate of *GAD/PV*[+]-*ChR2* cells (bottom). **(c)** PSTH of a putative excitatory L4 cell in response to repetitive whisker stimulation (red bars) in LED OFF and LED ON conditions (top) and population average spike count per stimulus of cells located in upper layers (bottom left) and deep layers (700–1,100 µM, bottom right). **(d)** Average whisker-evoked excitatory currents in two example L4 (left, top panel) and L5 (left, bottom panel) cells recorded independently during LED OFF and LED ON conditions and the population average currents (right). **(e)** Population depth profile of the thalamic contribution to evoked excitatory response of all recorded neurons (n = 47). **(f)** The contribution of thalamic input for each cell, when the principal whisker (PW) or adjacent whisker (AW) was stimulated. **(g)** Average EPSC in response to 10 mV step (left, top panel) and whisker deflection in two recorded cells during LED OFF and LED ON (left, bottom panel) conditions. Right panel shows population thalamic contribution versus change in input resistance. **(h)** PSTH of a ventral posteromedial nucleus (VPM) neuron in response to whisker stimulation during LED OFF (left, top panel) and LED ON (left, bottom panel) condition. Right panel shows the population average spike count per stimulus of VPM cells.

the reduction but not full loss of their response to whisker stimulation (Fig. 2b). We found that thalamic inputs could be substantially different for simultaneously recorded nearby cells (example in Fig. 2b). This was quantified by calculating the similarity index (SI) of their mean thalamic contributions (SI = $1 - \frac{|TC1 - TC2|}{TC1 + TC2}$, where TC1 and TC2 are the relative thalamic contributions of the two cells). The SI for the recorded pairs was only marginally higher than expected from individually recorded cells (Fig. 2c, computed using a bootstrap analysis, see the 'Methods' section), indicating that the large diversity in thalamic contribution in individual cells (Fig. 1e) was not the result of different experimental conditions.

**Ongoing cortical correlation is not driven by the thalamus.** During ongoing activity under anaesthesia, a prominent correlation in synaptic inputs of simultaneously recorded cells was

measured. The correlation coefficient (CC) between nearby L4 cortical cells during ongoing activity was comparable between current clamp and voltage clamp modes within the same pair (see example in Fig. 3a; mean CC = 0.5 ± 0.05 and 0.55 ± 0.04, respectively, P = 0.6875, n = 7 pairs, Wilcoxon signed-rank test). This implies that measurements under voltage clamp are a good estimate of the functional correlations between cells.

To reveal the contributions of thalamic inputs to cortical synchronized ongoing activity in L4, we compared the correlations between the excitatory synaptic currents in each pair when cortical firing was intact (LED OFF) to that calculated when cortical firing was silenced (LED ON). Although excitatory currents were highly synchronized in the intact cortex (Fig. 3b and Supplementary Figs 2 and 3; population mean CC = 0.39 ± 0.04, n = 10 pairs, Fig. 3c, upper panel, LED OFF), the correlation between the cells dropped substantially when we silenced cortical firing (Fig. 3b and Supplementary Figs 2 and 3;

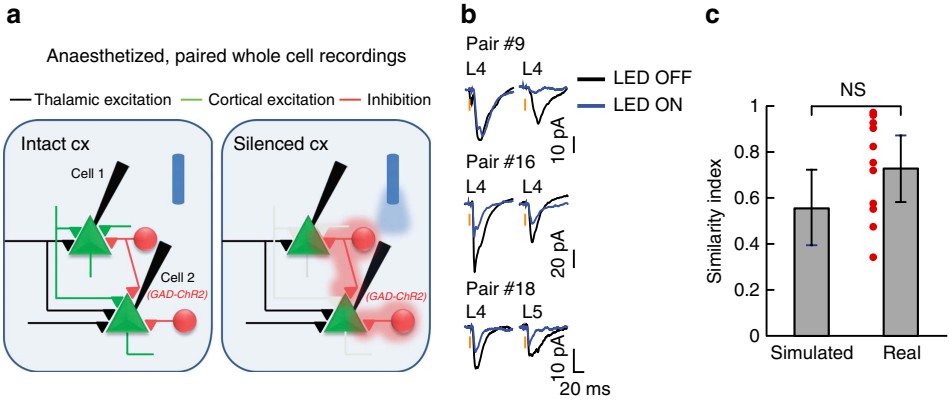

**Figure 2 | Thalamic contribution in nearby cortical cells is unequal.** (**a**) Schematic illustra1tion of paired whole cell recordings. (**b**) Average whisker-evoked excitatory currents in two example L4 cells recorded independently during LED ON and LED OFF conditions and below an example of an L4–L5 pair. (**c**) The similarity index (SI) of thalamic contribution ( ± s.e.m) for the 11 recorded pairs compared with the expected mean SI in simulated pairs from single cell recordings (bootstrap analysis, see the 'Methods' section).

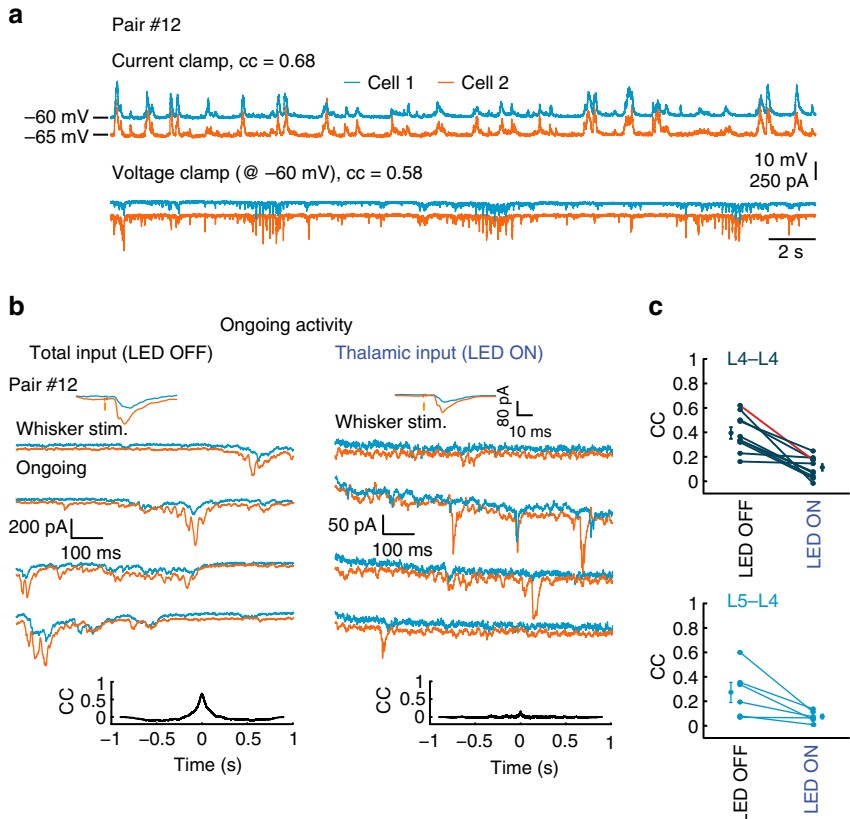

**Figure 3 | Isolated thalamic inputs exhibit weak correlation during ongoing activity.** (**a**) Simultaneous whole cell recordings of two putative excitatory L4 cells during ongoing activity in current clamp and voltage clamp (at the reversal potential of inhibition) modes recorded in anaesthetized mouse. (**b**) Example traces of ongoing excitatory currents in two simultaneously recorded L4 cells during LED OFF and LED ON conditions. Upper insets, average excitatory responses to whisker stimulation. Lower insets, cross-correlation between the cells. (**c**) Population CC values for L4–L4 pairs $n = 10$, $P = 0.002$, Wilcoxon signed-rank test) and L4–L5 pairs $n = 6$, $P = 0.0313$, Wilcoxon signed-rank test). Example pair in **b** is depicted in red.

population mean $CC = 0.11 \pm 0.02$, Fig. 3c, upper panel, LED ON, $z = -2.8031$, $P = 0.002$, Wilcoxon signed-rank test). We extended our database by recording from mixed pairs, where one cell was located in L4 and the other in L5 (Supplementary Fig. 4). Similar to L4–L4 pairs, CC of excitatory synaptic currents in L4–L5 pairs during ongoing activity (mean $CC = 0.27 \pm 0.08$, $n = 6$ pairs, Fig. 3c, lower panel, LED OFF) dropped significantly when cortical firing was silenced (mean $CC = 0.07 \pm 0.02$, Fig. 3c, lower panel, LED ON, $z = -1.862$, $P = 0.0313$, Wilcoxon

signed-rank test). The synaptic activity during LED ON decreased by $33 \pm 4.5\%$ ($P = 6e^{-6}$, Wilcoxon signed-rank test, measured from the total excitatory charge $Q$, the time integral of the EPSCs) and in an equivalent manner in both cells (Supplementary Fig. 5 shows $\frac{Q_{LEDon}}{Q_{LEDoff}}$ in each pair), indicating that reduced CC was not due to unequal light effects. Hence, synchronized ongoing activity in cortical neurons, within and across different thalamic recipient layers, depends on recurrent cortical activity and does not reflect the correlation between the cell's feedforward thalamic inputs.

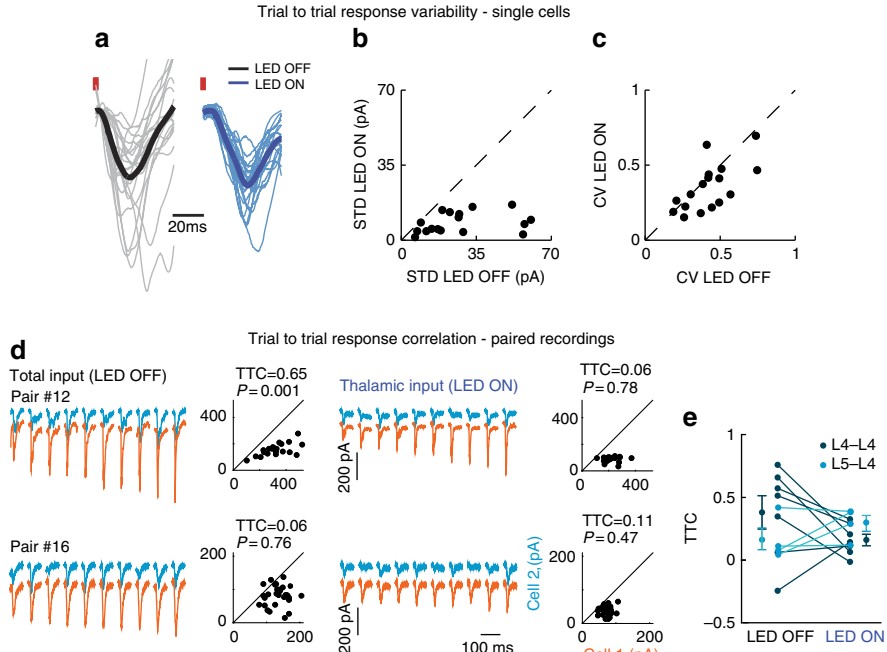

**Figure 4 | Sensory-evoked noise correlations are not determined by thalamic inputs.** (**a**) Example of all the responses of a representative L4 cell to whisker stimulation under LED OFF (grey traces with black trace for the mean) and under LED ON (blue traces with mean as dark blue) in anaesthetized animal. Data were normalized to the mean of each condition. (**b**) Population data of the standard deviation of the peak response measured for each trial at the two conditions. (**c**) Same as **b** for the coefficient of variation (CV). (**d**) $TTC_{EE}$ measurements during LED OFF (left) and LED ON trials (right). Responses to 10 sequential whisker stimuli of one cell (orange traces) sorted from the smallest to the largest amplitude with the corresponding responses of the second cell (blue traces). Scatter plots show peak excitatory current responses of one cell plotted against that of the second cell. (**e**) Population data and averaged $TTC_{EE}$ for LED OFF and LED ON conditions in L4–L4 pairs (mean $TTC_{EE} = 0.38 \pm 0.14$ and $0.16 \pm 0.05$, respectively, $P = 0.219$ Wilcoxon signed-rank test, $n = 7$) and in mixed L4–L5 pairs (mean $TTC_{EE} = 0.16 \pm 0.09$ and $0.29 \pm 0.06$, $P = 0.375$, Wilcoxon signed-rank test $n = 4$). TTC across the two conditions did not show significant correlation ($r^2 = 0.07$, $P = 0.557$ and $r^2 = 0.16$, $P = 0.596$ for L4–L4 and L4–L5 pairs, respectively).

**Cortical noise correlation is unrelated to thalamic inputs.** We next examined the cortical and thalamic contributions to trial-to-trial variability of whisker-evoked EPSCs of L4 cells. In the visual system, inactivation of cortical firing had a negligible effect on trial-to-trial membrane potential variability[4], suggesting that cortical variability is dominated by thalamic inputs. In contrast, we found that in the barrel cortex, optogenetic silencing of local firing profoundly reduced the variability of the whisker-evoked EPSCs (Fig. 4a–c). This was evident both in the standard deviation and in the coefficient of variation (CV). On average, the standard deviation of peak EPSPs (excitatory postsynaptic potentials) was reduced by $62\% \pm 22\%$ following cortical silencing (Fig. 4b, $P = 0.0003$, $n = 17$, $z = -3.6$, Wilcoxon signed-rank test). This trend remained even after normalization by the mean (Fig. 4c, reduction of $14\% \pm 29\%$ in CV, $P = 0.029$, $n = 17$, $z = -2.2$, Wilcoxon signed-rank test).

The larger trial-to-trial variability of intact cortex compared with inactivated cortex suggests that it is strongly influenced by recurrent cortical circuits. Therefore, we examined the cortical and thalamic origins of the sensory-evoked excitatory trial-to-trial correlation ($TTC_{EE}$, noise correlation) between pairs of cells that received direct thalamic inputs. Figure 4d shows two representative pairs in which 10 sequential whisker-evoked responses of one cell (orange traces) are sorted from the smallest to the largest, with the corresponding responses of the second cell (blue traces) during LED OFF (left) and LED ON (right) conditions (for illustration purposes, traces of the cell with the larger responses, on average, was placed below the second cell). Across the population, the change in $TTC_{EE}$ for the recorded pairs was not

consistent between the two conditions. In five pairs, $TTC_{EE}$ was reduced; in three, it roughly remained the same; and in two pairs, it was increased (Fig. 4e). No significant difference was found between the mean $TTC_{EE}$ values. For the seven L4–L4 pairs, the $TTC_{EE}$ dropped from $0.38 \pm 0.14$ to $0.16 \pm 0.09$ (Fig. 4e, dark blue circles, $z = -1.0142$, $P = 0.219$, Wilcoxon signed-rank test), and for four L4–L5 pairs, it changed from $0.16 \pm 0.05$ to $0.29 \pm 0.06$ (Fig. 4e, light blue circles $P = 0.375$, Wilcoxon signed-rank test). Notably, except for one pair (L4–L5), the $TTC_{EE}$ values for the two conditions were significantly different for all the pairs (bootstrap analysis of $TTC_{EE}$, see the 'Methods' section). Moreover, as the lines connecting the two conditions crossed each other, the $TTC_{EE}$ during LED OFF condition could not be predicted from the one measured during cortical silencing; that is, the distributions across the conditions were not correlated with each other ($r^2 = 0.07$, $P = 0.557$ and $r^2 = 0.16$, $P = 0.596$ for L4–L4 and L4–L5 pairs, respectively). Furthermore, noise correlation when cortex was intact, or during silencing, was not related to the ongoing correlations under the same conditions (Supplementary Fig. 6). Thus, similar to ongoing activity, we can conclude that the $TTC_{EE}$ of sensory response between cortical cells is not determined by direct thalamic inputs but rather depends on recurrent cortical activity.

**Origins of correlated cortical activities in awake animals.** Both ongoing and evoked activities in the cortical cells are regulated by animal behaviour[18,19,29,30,48]. Therefore, we wished to confirm our result in awake mice. Naive animals were head-fixed, and

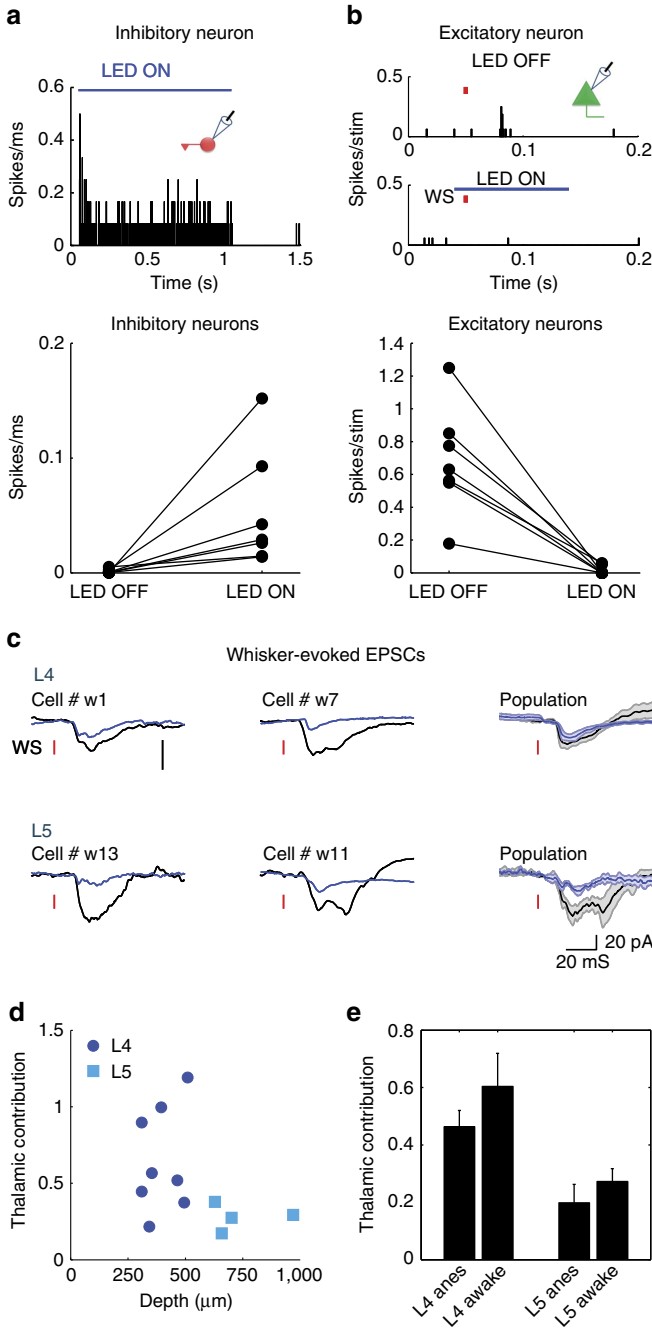

**Figure 5 | The contribution of thalamic input to whisker-evoked cortical response in awake mice.** (**a**) Peri-stimulus spike time histogram (PSTH) for an example GAD$^+$-ChR2 cell in response to 1s LED illumination (top) and population mean firing rate of GAD/PV$^+$ cells (bottom). (**b**) PSTH of a putative excitatory L4 cell in response to whisker stimulation (red bar) in LED OFF and LED ON conditions (top) and population average spike count per stimulus of cells located in layers 4 and 5. (**c**) Average whisker-evoked EPSCs of two examples L4 (top) and L5 (bottom) cells recorded independently during LED OFF and LED ON conditions. To the right are the average EPSCs of the two populations. The two vertical scale bars represent 20 pA. (**d**) Population depth profile of the thalamic contribution to the EPSCs of all recorded neurons ($n = 12$). (**e**) Mean population data of the thalamic contributions in L4 and L5 of awake and anaesthetized mice.

were not trained to perform a task. Similar to our recordings in anaesthetized animals, surface illumination of the cortex activated GABAergic cells (Fig. 5a, example cell and population data below,

$1.3 \pm 0.7$ versus $47 \pm 16$ Hz for LED OFF and LED ON; $n = 7$ cells, $z = -2.63$, $P = 0.015$, Wilcoxon signed-rank test). This in turn inhibited the local circuitry and blocked almost completely (98%) whisker-evoked firing of the non-ChR2 expressing cells at all depths (example of a L4 cell is shown in Fig. 5b and the population data below, $0.7 \pm 0.1$ versus $0.02 \pm 0.01$ spikes per stimulus for LED OFF and LED ON; $n = 7$ cells, $z = -2.63$, $P = 0.015$, Wilcoxon signed-rank test).

To estimate the contribution of thalamic input to the total whisker-evoked excitatory current in awake mice, we recorded from 12 neurons located at L4 and L5. The average excitatory currents in four cells are demonstrated during intact cortical activity (Fig. 5c, black traces) and when cortical firing was silenced (Fig. 5c, blue traces). Similar to anaesthetized mice, a large variability in the contribution of thalamic input was also found in awake mice and as before, the thalamic contribution in L4 cells was larger compared with L5 cells (Fig. 5d, $0.59 \pm 0.1$ and $0.27 \pm 0.04$, respectively, $P = 0.02$, unpaired $t$-test). Importantly, we found no differences between awake and anaesthetized mice when comparing the thalamic contribution in each layer (Fig. 5e, $P = 0.275$, two-way analysis of variance).

Whole cell recordings in awake animals were made simultaneously with LFP recordings using an additional glass pipette that was placed in proximity to the recorded cell in L4 (Fig. 6a, $< 200 \, \mu m$). Inactivation of cortical firing reduced the amplitude of the averaged LFP response to whisker stimulation (Fig. 6b, two examples of simultaneous cell-LFP recordings). Population analysis showed that thalamic contribution varied also for LFP recordings (Fig. 6c). On average, the contribution of thalamic inputs to the two signals was similar (Fig. 6c, $0.57 \pm 0.1$ versus $0.46 \pm 0.1$, $P = 0.57$, $z = -0.56$, Wilcoxon signed-rank test), showing that as for the EPSCs, a significant amount of the evoked LFP response arises from thalamic input.

To reveal the contribution of shared thalamic inputs to cortical synchrony during ongoing activity in awake mice, we calculated the correlation between membrane potential and the LFP when cortical firing was intact and when firing was optogenetically silenced. Importantly, we found that the magnitude of the correlation coefficient (CC) between the LFP signal and the electrical activity of the recorded cells was independent of the recording mode method (Fig. 6d, $CC = -0.45$ at current clamp and $CC = 0.47$ at voltage clamp). Hence, voltage-clamp recordings capture the functional correlations that exist between subthreshold activity of individual cells and the LFP signal. We interleaved trials in which cortical firing was intact with trials in which cortical firing was silenced and calculated the correlation between EPSCs of L4 cells and the LFP signals during ongoing activity. The example paired recording in Fig. 6e shows that the thalamic contribution for both signals is nearly one, as evident from the negligible change in the average responses of both signals to whisker stimulation when the cortex was illuminated (Fig. 6e, inserts). Yet, optogenetic silencing of cortical firing during ongoing activity drastically reduced the correlation between the excitatory current of the recorded cell and the nearby LFP signal (from 0.68 to 0.13). Silencing cortical firing in seven similar recordings reduced the correlation between the LFP and the excitatory current during ongoing activity from $0.31 \pm 0.08$ to $0.03 \pm 0.03$ (Fig. 6f; $n = 8$, $P = 0.01$, $z = -2.8$, Wilcoxon signed-rank test). We can, therefore, conclude that cortical synchrony during ongoing activity in L4 of the barrel cortex in awake mice is not driven by thalamic inputs.

Next, we examined the effect of cortical silencing on the trial-to-trial correlation between intracellular excitatory current and a nearby LFP signal in response to vibrissa stimulation (TTC$_{EL}$). Similar to anaesthetized mice, cortical silencing reduced the response variability of the individual cells and of the LFP

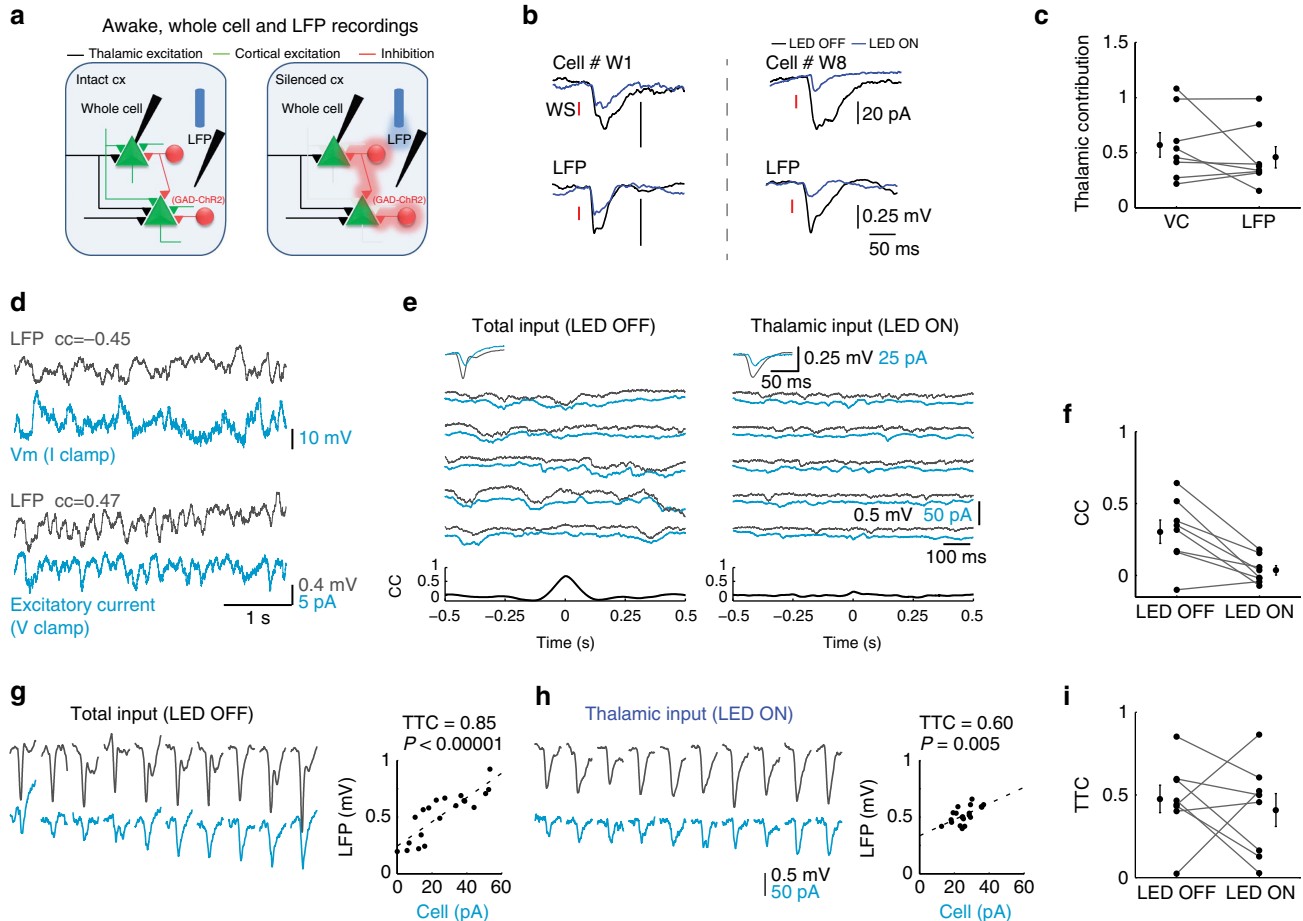

**Figure 6 | Ongoing and whisker-evoked correlations in awake mice are not determined by thalamic inputs.** (**a**) Schematic illustration of whole cell and LFP paired recordings in awake mice. (**b**) Average of paired recordings showing mean whisker-evoked EPSCs and LFP responses recorded from two animals. (**c**) Population data and mean peak EPSCs and the corresponding LFP response under LED OFF and LED ON conditions. (**d**) Simultaneous whole cell and LFP recordings in L4 during ongoing activity when the cell was recorded in current clamp and voltage clamp (at the reversal potential of inhibition) modes. (**e**) Example traces of ongoing excitatory currents in a L4 cell and the corresponding nearby LFP signal that was recorded simultaneously during LED OFF and LED ON conditions. Upper insets, average whisker-evoked EPSCs and LFP under the same conditions. Lower insets, cross-correlation between the cell and the LFP. (**f**) Population CC values for L4–LFP pairs $n = 8$, $P = 0.01$, Wilcoxon signed rank test). (**g,h**) $TTC_{EL}$ of EPSC–LFP paired recordings during LED OFF (**g**) and LED ON (**h**) conditions. Responses to 10 sequential whisker stimuli of one cell (light blue traces) sorted from the smallest to the largest amplitude with the corresponding responses of LFP (black traces). Scatter plots show peak LFP responses against the peak EPSC response of the cell. (**i**) Population data and averaged $TTC_{EL}$ for LED OFF and LED ON conditions in EPSP–LFP paired recordings ($TTC_{EL} = 0.47 \pm 0.1$ and $0.41 \pm 0.1$, respectively, $P = 0.57$ Wilcoxon signed-rank test, $n = 7$).

(Supplementary Fig. 7), suggesting that noise correlations are affected by recurrent cortical circuits. Indeed, the example in Fig. 6g shows that the response to vibrissa stimulation varied considerably from trial to trial for both signals. Sorting the cellular responses from the smallest to the largest, while plotting the corresponding LFP signal, reveals a clear correlation between the two signals ($r^2 = 0.85$, $P = 0.00001$). Silencing cortical firing by light reduced the $TTC_{EL}$ (Fig. 6h to $r^2 = 0.6$ $P = 0.005$). On average, no significant change in $TTC_{EL}$ was found between the two conditions (Fig. 6i, $TTC_{EL} = 0.47 \pm 0.1$ and $0.41 \pm 0.1$ for LED OFF and LED ON, respectively, $P = 0.57$, $z = -0.56$, $n = 7$ Wilcoxon signed-rank test). Importantly, the $TTC_{EL}$ values for the two conditions were significantly different for all the pairs (bootstrap analysis of $TTC_{EL}$, see the 'Methods' section). As in anaesthetized animals, the $TTC_{EL}$ in the intact cortex was not correlated to that found during cortical silencing ($r^2 = 10^{-5}$, $P = 0.99$). Taken together, noise correlation between single cell and the population LFP response in the intact cortex cannot be inferred from the measured correlations of the isolated thalamic

inputs, implying that cortical recurrent connections determine the degree of synchronization in L4 of awake mice.

## Discussion

In this work, we investigated the role of thalamic inputs in shaping the synaptic correlations between neighbouring cells in thalamic recipient cortical layers during ongoing and sensory-evoked activities. To address this question, we optogenetically silenced cortical firing in anaesthetized and awake mice to isolate the thalamic excitatory inputs of intracellularly recorded cells in L4 and L5. Specifically, in awake mice, we examined the effect of silencing cortical firing on the correlation between excitatory inputs of individually recorded cortical cells and a nearby LFP signal, whereas in anaesthetized mice, we simultaneously recorded the excitatory inputs of nearby pairs of neurons. Our results show that synchronized activity during ongoing activity emerges from intracortical inputs, rather than being driven by direct thalamic inputs. Trial-to-trial sensory-evoked correlation

('noise correlation') in response to vibrissa stimulation during intact cortical firing is poorly related to the noise correlation in the thalamic inputs, indicating that it is also a product of intracortical recurrent activity.

To isolate the thalamic input of cortical cells, we illuminated the barrel cortex of transgentic mice expressing *ChR2* in *GAD+* cells while recording excitatory currents. A similar approach was used in recent studies of the auditory and visual cortices of anaesthetized mice[38–40] where ChR2 was expressed in PV+ cells. Owing to the high level of expression in these transgenic mice, it is reasonable to assume that neurons that did not exhibit direct activation by light are excitatory cells. In anaesthetized mice, we found that thalamic inputs contributed on average about 46% of the total excitatory input of layer 4 cells, which is slightly higher than previously reported in the visual cortex ($\sim$30%; ref. 38) and roughly the same as in the auditory cortex of mice (41%; ref. 40). Our estimate is slightly lower than in the rat auditory cortex where cortical firing was pharmacologically silenced (61%; ref. 49). The higher contribution of thalamic inputs to the total response of L4 neurons in the barrel cortex, compared with the visual cortex, may reflect the stronger damping of recurrent cortical activity owing to the prominent feedforward inhibition in the somatosensory[38,50,51] and auditory[20] cortices. Thus, across different modalities, recurrent cortical circuits may amplify thalamic inputs in a slightly different manner. Bruno and Sakmann[37] suggested, however, that cortical amplification is not required to explain the sensory response of L4 cells in the barrel cortex. In a few cells, we indeed observed almost no amplification (thalamic contribution $\sim$100%), but for most cells, amplification was prominent. It is possible that this discrepancy reflects differences across species (mice in our study and rats in ref. 37).

To the best of our knowledge, our study is the first that compared the amplification of thalamic inputs by recurrent cortical circuits across anaesthetized and awake mice. In both conditions, the contribution of thalamic input to the total excitatory input varied with the recording depth. Higher contribution of thalamic input was found in cells that were recorded in L4 compared with L5 cells. The higher contribution of thalamic inputs in L4 is expected from the dense innervation of L4 by thalamic inputs[41,44,52]. Importantly, the contribution of thalamic input in L4 and L5 of anaesthetized mice was very similar to the contribution of these inputs to the same layers in awake mice.

Similar to previous studies of the visual and auditory cortices[38–40], cortical silencing showed that the contribution of thalamic inputs was invariant to the optimality of stimulation. We demonstrated it comparing the contribution of thalamic inputs when independently stimulating the PW and one of the AWs within the same cells. This suggests that the local circuitry of a cortical cell amplifies its thalamic inputs in a particular manner for each cell, regardless of the feature that activates this cell.

Recurrent cortical activity engaged quite rapidly, roughly at the same time of the onset of thalamic input, as evident when the average response to whisker stimulation in intact cortex was compared with the time course of thalamic input alone under cortical silencing. Naively, one would expect that recurrent inputs will be delayed by a few milliseconds with respect to the onset of thalamic input, as it involves at least one additional synapse. However, this does not necessarily need to be the case. Assuming that we sampled the cortical population randomly, the thalamic input would arrive to some cells relatively early, whereas to others it would arrive later, The earliest firing in L4 can be as short as 5 ms following whisker stimulation[53], therefore, local cortical cells should provide input to other cells roughly at the same time or even before the onset of thalamic input of cells that are not the 'primer' cells. Indeed, the latency of the response under cortical silencing in some of our recorded cells was clearly delayed by a couple of milliseconds relative to the onset of the response when the cortex was intact (Fig. 6b, cell #W8). Hence, recurrent cortical activity engages rapidly in the somatosensory cortex. Rapid amplification of thalamic inputs is evident also in the studies of the visual and auditory cortices[38,40].

The origin of noise correlation of sensory responses in primary sensory cortical areas is under dispute. In one view, the variability in sensory responses mostly depends on subcortical processing of sensory inputs and therefore on the variability of the thalamic inputs. According to this possibility, cortical responses fluctuate from trial to trial due to noisy thalamic inputs, and since these inputs are shared, the responses of different cortical cells are correlated. This view is supported by measurements of trial-to-trial spiking variability in geniculate cells of the visual system[54] and the similarity in the variance of membrane potential responses to brief visual stimulation of L4 cells in V1, before and after inactivation of cortical firing using electrical stimulation[4]. However, our results are in line with a different view, suggesting that cortical synchrony emerges due to intracortical inputs[5,9,25–28]. In contrast to the findings of Sadagopan and Ferster[4], our experiments show that the cortex adds substantial variance to that which originates from the thalamic inputs, as the standard deviation of the response was significantly larger during intact cortical firing compared with that measured when the cortex was silenced. Direct measurements of trial-to-trial correlation indicate that they are not determined by thalamic inputs. In awake mice, the degree of correlation between excitatory inputs of L4 cells and nearby LFP signal, when the cortex was intact, was highly variable across the population. This was observed both for ongoing activity (Fig. 6f) and for the evoked response (Fig. 6i). A large range of correlations was also measured between the excitatory currents in anaesthetized mice. In other studies of the barrel cortex in awake mice[18,19], nearby cells, recorded in layer 2/3, exhibited a much narrower range of correlation during ongoing activity and on average it was higher than what we report in this study of L4. This discrepancy probably reflects laminar differences in the strength of correlations. Indeed, the correlated variability of extracellularly recorded neurons in upper cortical layers of the visual cortex of awake monkeys was found to be significantly higher than between cells located in the granular layer (that is, layer 4)[55].

The wide range of correlation strengths between individually recorded cells and the nearby LFP in awake animals, as well as between the paired intracellular recordings in anaesthetized animals, is reminiscent of the large diversity of network coupling that was recently reported in the visual cortex using extracellular recordings[56]. According to the study of Okun and his colleagues, the diversity in network coupling is related to the strength of synaptic connections made between individual cells and their neighbouring population. They showed that cells that received more synaptic inputs from their neighbours exhibited higher coupling with network activity[56]. Although we have no direct evidence that supports this conjuncture in our study, we know from our recordings that the strength of ongoing and noise correlations between cortical cells are weakly dependent on their shared thalamic inputs. This happens despite the large contribution of thalamic inputs to the total sensory-evoked excitatory currents ($\sim$50%).

Several factors contribute to noise correlation of thalamic excitatory inputs of cortical cells upon sensory stimulation. These include noise correlation of thalamic cells, the convergent–divergent organization of thalamo-cortical ascending axons and co-modulation of thalamo-cortical synapses by neuromodulators or GABA(B) presynaptic contacts[46]. In addition, the reliability of

axonal transmission of individual thalamic cells, such as result from axonal failure and from the reliability of synaptic release, may reduce the noise correlation between cortical cells. Because of the large number of thalamic fibres that converge on each cortical cell[37,44], due to averaging, the reliability in axonal conductance or synaptic release should have a negligible effect on the variability of the total synaptic thalamic input of individual cortical cells or their impact on correlated variability. Our method, however, bypasses these factors all together, allowing us to measure the impact of these factors when summed together.

In conclusion, we found that synaptic correlations of nearby cortical cells in L4 and L5 during ongoing and sensory-evoked activities are poorly related to their thalamic excitatory inputs. Moreover, the contribution of thalamic inputs varies considerably across the population. The functional role of the asynchronous nature of thalamic inputs and diversity in thalamic contribution in L4 is unclear. An intriguing possibility is that such connectivity may smooth the population response curve to a wide range of stimuli, allowing better encoding of sensory inputs. Indeed, the amount of possible information that could be extracted from a neuronal population is constrained by the noise correlations of the response[57]. Therefore, emergence of synchronous activity locally by cortical mechanisms is essential for cortical computations.

## Methods

**Animals.** All the procedures involving animals were reviewed and approved by the Weizmann Institutional Animals Care Committee. Recordings were made on young adult mice of either sex (9–16 weeks old) housed up to five in a cage with a 12/12h dark/light cycle. Two strains were used, *GAD-CRE* mice (JAX #010802) and *PV_CRE* mice (JAX #008069) crossed with a *ChR2* reporter strain (JAX #012569). Since cortical firing was silenced similarly in both strains, data were pooled from both types.

**Anaesthetized animal preparation.** For intracellular recording from the barrel cortex, after initial anaesthesia with ketamine (90 mg kg$^{-1}$; intraperitoneal) and xylazine (2 mg kg$^{-1}$; intraperitoneal), a tracheotomy was made and the animals were mounted in a stereotaxic device and artificially respirated with a mixture of halothane (0.5–1%) and oxygen-enriched air. The scalp and fascia were removed and a metal headplate was mounted over the left hemisphere using dental cement (Lang dental) and VetBond (3 M). A craniotomy (∼1 mm in diameter) was made above the barrel cortex (centred 1.3 mm posterior and 3.3 mm lateral to the bregma) and a portion of the dura mater was carefully removed. The craniotomy was constantly washed with artificial cerebrospinal fluid containing (in mM): 124 NaCl, 26 NaHCO$_3$, 10 glucose, 3 KCl, 1.24 KH$_2$PO$_4$, 1.3 MgSO$_4$ and 2.4 CaCl$_2$. The levels of end-tidal CO$_2$ and heart rate (250–450 beats per minute) were monitored throughout the experiments. Body temperature was kept at 37 °C using a heating blanket and a rectal thermometer.

**Awake animal preparation.** Animals underwent the implantation of a head bar to allow awake head-fixed recordings as follows: following initial anaesthesia in an induction chamber containing a mix of isoflurane and oxygen-enriched air, the animals were mounted in a stereotaxic device, and kept deeply anaesthetized, monitored by checking for lack of reflexes and pace of breathing. The area of incision was treated with lidocaine and cleaned with iodine and 70% ethanol. The skullcap was exposed and cleaned. The skull above the barrel cortex (1.3 mm posterior, 3.3 mm lateral to the bregma) was covered with silicon glue (Smooth-On, Inc., USA). A small titanium headbar was firmly affixed to the skull slightly anterior to bregma with dental acrylic (3M, Germany).

Following a recovery period (4–7 days), the animals were anaesthetized in an induction chamber containing a mix of isoflurane and oxygen-enriched air, the animals were then mounted in a stereotaxic device and kept deeply anaesthetized. The silicon glue covering the skull over the barrel cortex was removed and a craniotomy was performed exposing the barrel cortex and leaving the dura intact. The brain was then covered in an agar layer (2% w/v) held in place with silicon glue and the animal was returned to the cage for a recovery period (1–2 h). The animal was then returned to the set and head-fixed for the electrophysiological recordings.

**Cortical patch recordings.** Borosilicate micropipettes were pulled to produce electrodes with a resistance of 4–10 MΩ when filled with an intracellular solution containing the following (in mM): 135 Cesium-Met., 4 TEA-Cl, 10 HEPES, 1 MgATP, 0.3 NaGTP, 3 QX-314 and 10 phosphocreatine (310 mOsm). Intracellular signals were acquired using an Axoclamp-700B amplifier (Molecular Devices) and

low passed at 3 kHz before being digitized at 10 kHz. Recording depth ranged between 300 and 700 µm. The cells up to 500 µm were classified as layer 4 cells based on their depth and the response latency (Supplementary Fig. 8), while cells recorded from this depth and up to 700 µM were classified as layer 5 cells. For double intracellular recording, two patch pipettes were inserted into the brain upto a depth of 300 µm. After reaching a successful whole cell recording in one pipette, the second pipette was advanced until the second whole cell recording was obtained.

Voltage clamp recordings were started immediately after a successful breach of a giga-seal. To record only excitatory currents under voltage clamp, membrane potential was clamped at the reversal potential of inhibition. This potential was determined under voltage clamp for each cell by adjusting the holding potential until no change in current was measured upon activation of the *GAD* or *PV ChR2* with LED illumination. Current-clamp recordings were also made in a subset of cells.

For simultaneous LFP recordings a patch pipette was inserted to a recording depth of 400 µM. The signal was band passed at 0.1–300 Hz before being digitized at 10 kHz.

**Thalamic extracellular recordings.** Extracellular recordings were performed using Juxta electrodes filled with patch solution with a resistance of 20–30 MΩ. The craniotomy was centred 1.5 mm lateral and 1.5 mm posterior of the bregma over the ventral posteromedial nucleus at a depth of 3.6 mm. The signals were amplified using Axoclamp-700B amplifier, low passed at 3 kHz and digitized at 10 kHz.

**Cortical silencing.** To activate *ChR2*, an LED light source at 460 nm (Prizmatix Opt-LED-460) was coupled to a bare optical fibre (200 µm diameter, 0.22NA; ThorLabs M25L05) placed above the cortex. The LED was driven by an analogue output from our acquisition system (National Instruments) for 1 s. The intensity of the light was around 7 mW at the tip of the fibre. As in Liu *et al.*[49], we estimated the effect of cortical silencing on the inputs resistance of the cells. A step current of ± 100 pA was injected under LED OFF and LED ON conditions. Cortical silencing decreased input resistance (from 330 ± 25 to 220 ± 20 MΩ, $P = 0.006$, two-tailed paired Wilcoxon signed-rank test, $n = 15$). We estimated how much the decrease of input resistance would affect the recorded current amplitude based on:

$$I_{rec} = \frac{R_{in}}{R_{in} + R_s} \times I_{syn}$$

where $I_{syn}$ is the actual amplitude of synaptic current, $I_{rec}$ is the recorded amplitude, $R_{in}$ is the input resistance and $R_s$ is the effective series resistance. $R_s$ was unchanged after cortical silencing (70 ± 25 to 60 ± 20 LED OFF versus LED ON; $P = 0.68$, two-tailed paired Wilcoxon signed-rank test, $n = 6$). Assuming no change in $I_{syn}$, the decrease in $R_{in}$ and $R_s$ during illumination would lead on average to a 5% reduction of the recorded synaptic amplitude, which is negligible compared with the measured amplitude reduction after cortical silencing.

**Whisker stimulation and protocols.** Whiskers were trimmed to a length of 10–20 mm. When single whisker stimulation was given, either the PW or AW were inserted into 21G needle attached to a galvanometer servo motor with a matching servo driver and controller (6210H, MicroMax 677xx, Cambridge Technology Inc.). The displacement was measured off-line using an optical displacement measuring system (optoNCDT 1605, Micro-Epsilon), indicating that ringing was negligible. A fast-rising voltage command was used to evoke a fast whisker deflection with a constant rise time of ∼1 ms followed by a 20 ms ramp-down signal. The stimulation velocity and the corresponding deflection amplitude (∼50 mm s$^{-1}$, 145 µm amplitude) were adjusted to evoke clear subthreshold responses in the cortical cells. When global whisker stimulation was used, the tip of the needle was placed on the whisker pad and multiple whiskers were stimulated simultaneously.

Whisker stimulation was delivered without and with LED illumination, which started 300 ms before the whisker was stimulated and the light was turned on for 1 s. Trials with LED stimulation alone were also delivered and they were used to correct drifts in voltage recordings, if occurred. These trials were pseudo-randomly delivered and were either 3 or 5 s long with 2 s inter-trial intervals. Each condition was repeated at list six times.

**Data analysis.** The recordings were analysed using custom software written in MATLAB (The MathWorks). We smoothed the raw traces using a symmetric Savitzky–Golay filter with a first-order polynomial and a window size of 21 points. The amplitude of the EPSC was measured as the difference between the minimum peak membrane current response and the mean baseline value obtained over 10 ms before stimulation. Excitatory charge (Q) was calculated as the time integral of the EPSCs over a period of 900 ms before or during LED illumination. Spike counts were calculated as the sum of spikes observed during the 5–45 ms period after whisker stimulation without subtracting spontaneous firing, because in the absence of stimulation, cells fired very sparsely. Crosscorrelation coefficient (CC) between each paired traces (at 0 time-lag) was calculated and the average CC between shuffled traces was subtracted from this value. Data are presented as mean ± s.e.m.

Statistical difference between the thalamic contribution in the recorded pairs was calculated using bootstrap analysis where traces from LED ON and OFF were pooled randomly and averaged for each cell in the pair. The thalamic fraction was calculated 300 times, averaged and compared (Wilcoxon rank-sum test) to that calculated for the second cell in the pair. Similarity index (SI) of cortical amplification was calculated as follow: $= 1 - \frac{|TC1 - TC2|}{TC1 + TC2}$, where TC1 and TC2 are the relative thalamic contributions of the cells. $SI = 1$ implies that thalamic contributions are identical. The mean SI of simulated pairs was calculated using bootstrap analysis by artificially constructing 11 random pairs (the same number of pairs in our data base) from the individually recorded neurons and repeating this procedures 500 times to obtain mean and 95% confidence limits. Significant changes in trial-to-trial correlations between LED OFF and LED ON conditions (Figs 4e and 6i) were computed using bootstrap method (100 repetitions), where for each repetition, we computed the correlation between a random number of events, as the number of trials for that pair, with repetitions. The resulting populations were compared using Wilcoxon rank-sum test.

**Data availability.** The data that support the findings of this study are available from the corresponding author on request.

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

## Acknowledgements

This work was supported by ISF, Minerva and DFG-SFB 1089. We thank all the members of the I.L. laboratory and especially Y. Katz, I. Meir and R. Tumasus for their helpful contributions. We acknowledge M. Carandini and M. Okun for their thoughtful comments on the manuscript.

## Author contributions

K.C.-K.M. and B.M. contributed equally to this work. K.C.-K.M., B.M and I.L. designed the experiments; K.C.-K.M. performed the experiments with A.N.R and B.M., K.C.-K.M., B.M. and I.L. analysed the data; K.C.-K.M., B.M. and I.L. discussed the results and wrote the manuscript.

## Additional information

**Competing financial interests:** The authors declare no competing financial interests.

