## [Peer Review File · Nature Communications]

Transferred manuscripts:

Reviewer #1:

1. The manuscript investigates the role of local circuit cortical activity in generating correlations in the cortex compared to synchronous activity due to thalamic inputs. The question of how information is encoded across large assemblies of neurons, the nature of the neural code, is tightly linked to the question of neural co-variability. Despite the usefulness of the data provided in the current work the significance of the question addressed is not very clear. For a general interest journal like *Neuron* the authors need to do a better job with additional analysis and experiments to put their questions and results in a more general framework. On the data analysis side the main concern is that the experiments were done in anesthetized animals. Recent work by many labs (e.g. Petersen) have underscored the importance of different internal brain states in generating correlated activity in cortical circuits and the importance of precisely defining the brain state (e.g. exploratory, quiet wakefulness including monitoring of pupil etc). The authors need to redo their experiments in awake animals while monitoring brain state and condition in different brain states. This will add a critical dimension to their work increasing its significance.

This is an excellent comment and we have added a new line of experiments in which we investigated this question in awake mice. In the anesthetized mice, we have used paired intracellular recordings. In awake animals, we recorded excitatory currents of L4 cells simultaneously with LFP using a nearby electrode. As a result of this major change we decided to change the format of the manuscript from a brief communication to a full article. The results that we obtained in awake animals

strongly support the anesthetized data. Namely, the correlations of ongoing and sensory evoked activities in layer4 do not result from thalamic input but rather result from intracortical recurrent inputs. We include two new figures that describe these results (figure 5 and 6).

2. Moreover, I did not find that the paper was clearly written, especially the second half (i.e. description of Figure 2 and Figure 3). The first half about silencing cortical activity reduces correlation of L4 cells is relatively convincing.

Due to the nature of Brief Communications it is possible that our paper was not clearly written. We believe that this version is clearer.

3. A common issue of this type of manipulation of shutting down the cortex is that it does not only blocks cortical activity, but also changes the corticothalamic feedback loop, which influences the thalamic inputs to L4 neurons. The authors should discuss this in detail or address with additional experiments. How do their conclusions depend on this fact?

This is an excellent comment. We made additional experiments and recorded firing in 9 thalamic cells. These experiments show that cortical silencing had no effect on their response to whisker deflection. This data is now presented in Figure 1h.

4. As discussed above the experiments were performed under anesthesia. What happens if they are performed in awake animals? During wakefulness, mice have multiple microstates. Is the synchronous period during the wakefulness still due to the cortical activity or does thalamic input play a role in this case?

As stated above, we performed EPSC-LFP simultaneous recordings in awake animals that strongly support our conclusions. Figures 5 and 6 are dedicated to this part.

5. I found it rather surprising that the CC of spontaneous activity does not predict at all the TTC.

We are also puzzled by these results. It probably means that the circuits that determine the CC do not fully overlap with the circuits that determine the TTC.

The number of pairs in Figure 2c-e are too low to draw a convincing conclusion and more data are necessary.

As stated above, we have additional recordings in awake animals that strongly support the data obtained from anesthetized mice and our conclusions.

6. In Figure 2c, pairs of L4-L4, 5 cells clearly have higher TTC with LED off than LED on. With more data points, it seems that TTC with LED off could be higher than TTC with LED on.

Although increasing our dataset might result in the conclusion that there is a significance difference between TTC during LED OFF and LED ON, it would not change the conclusion that one cannot predict the TTC in the intact brain from the TTC of thalamic inputs. However, even after polling our awake and anesthetized results, there is no significant difference between the means in the two conditions, and as before the values of TTCs across the two conditions are not correlated, indicating that one cannot predict the TTC in the intact brain from the TTC of the thalamic inputs.

7. Another possibility is that the pairs shown in Figure 2c contains two groups. One group of pairs have higher TTC in the LED off condition and the other group has the opposite effect. The difference between the two groups could be whether the pair of the cells are connected together or not. This should be tested.

This is a very interesting possibility that will be very hard to test using our dataset. We feel that splitting the data according to the effect we see in each pair will not reveal any additional information. Also, as we have not checked for connectivity in our anesthetized recordings due to the filling of the recording pipettes with qx-314. In awake mice we cannot test the connectivity since one of the recording sites was an LFP recording. Nevertheless, we feel that this question is beyond the scope of this manuscript.

8. How deep in the brain does the LED light can reach? The authors should also test neural activity of cells in different layers to measure their effect of silencing as a function of depth. If neurons in deep layers are not silenced, then the results should be reinterpreted.

This is a very good question. To answer this, we performed additional experiments using cell attach recordings from putative excitatory cells (i.e, were not activated by blue light) situated in deep layers (700-1100 μ M depth). We show that indeed their response to whisker stimulation was silenced by the activation of blue light at the cortical surface. This new data was added to the article in Figure 1c, bottom right panel.

9. Figure 3c-d, are those results include both L4 and L5 cells? Can the authors color-code them?

The groups are color coded.

10. Why the thalamic contribution of #19 in Figure 3g is greater than 1?

This is the data for this cell. For this cell the average EPSP during cortical inactivation was larger than measured when the cortex was intact.

11. Figure 3d is not clearly motivated in the text or the legend.

As mentioned above we rearranged the manuscript in a way that we believe will more clearly motivate this figure.

Reviewer #2:

The main goal of the this manuscript was to determine the role of thalamic input in correlating subthreshold activity in L4 neurons. The authors show that silencing the local cortical network by activating interneurons expressing chr2 with blue light leads to desynchronization of the subthreshold activity of L4 cells. I find more problems and inconsistencies than merits in this study.

1. The reasoning is somehow circular: the goal is to understand the role of thalamic input to the ongoing activity but the manipulation eliminates entirely the ongoing activity, leaving only a small amount of presumably thalamic input with very little correlation.

We disagree with the reviewer's claim that this is circular reasoning. That the ongoing activity was reduced and became uncorrelated is a strong indication that these inputs do not contribute to the correlated ongoing cortical activity. Moreover, one could claim that a small amount of thalamic input is amplified in a consistent manner that keeps the correlation that exists in these remote inputs. But our results clearly show that this is not the case. If correlated activity in the cortex originated

from thalamic inputs we would clearly see it in the remaining inputs. In summary, we argue that there is no circularity in our experimental design or conclusions.

2. Furthermore, thalamic input seems to engage very variable amounts of cortical activity and therefore, it seems possible that cortical synchrony could be driven entirely by a few stronger thalamic inputs amplified by the cortical network. But this is a very different conclusion from that reached by the authors, i.e., that the intracortical synchrony is due entirely to cortical mechanisms.

The hypothesis that the reviewer suggested is very interesting and could be correct. Theoretically, synchrony could be driven by few strong thalamic inputs. Yet, since in all the pairs ongoing synchrony was reduced following silencing it is clear that intracortical firing is essential for the generation of cortical synchrony.

3. It is obvious that without the reverse experiment, in which thalamus is silenced, the answer to the question of what amount of synchrony is due to intracortical mechanism remains unknown. On the other hand it is well known that cortical synchrony during slow oscillatory activity is preserved after extensive thalamic lesions, a fact not mentioned by the authors and which would be trivially recapitulated by their data.

Although the proposed experiment is trivial, it clearly cannot be used when studying the sensory evoked correlations in the cortex, as such procedure will abolish the response to sensory stimulation due to thalamic inactivation. As the proposed experiment is irrelevant to our central question, we decided to use the same approach when studying the origin of ongoing and sensory evoked correlations. Using the same approach allows us to compare how correlations are affected in the same pair of cells in both conditions. Furthermore, although blocking thalamic activity is likely to

reduce cortical activity (Poulet et al. nature neurosci., 2012) it is well possible that a low level of synchronized thalamic inputs are amplified and manifested in the correlation that we see in the cortex. The above citation was added in the introduction.

4. Since there is no quantification of the state of the animal, and indeed the two examples in Figure 1 seem very different states, it is very hard to interpret the results of this study

It is unclear to us what the reviewer means when mentioning two states. All the voltage and current recordings in Figure 1 were taken from the same recorded pair (pair no. 12) under the same conditions. Importantly, as mentioned to reviewer 1, we have also added a significant amount of data from awake animals that are in agreement with our conclusions from anesthetized animals.

5. I don't see how the authors can conclude that the thalamic inputs were not affected by the optogenetic manipulation (bottom page 4) since thalamic inputs cannot be measured before turning on the blue light.

If we understand the reviewer's concern, he/she is worried that thalamic firing was affected by the light. However, in the manuscript we argue that the firing of thalamic cells was not affected by cortical silencing. We have extensively extended our control experiment in which the sensory response of thalamic cells was recorded with and without cortical silencing (see fig. 1h). In addition, we show in Supplementary Figure 1 that the effect of cortical inactivation on excitatory response is immediate, indicating that the attenuation was not due to slow activation of presynaptic GABA(B) receptors.

6. Please show or explain measurements of inhibitory reversal potential used to set the V_m in cc and vc.

For VC recordings the reversal potential of inhibition for each cell was measured under voltage clamp while activating inhibitory cells by blue light (LED ON). Light induced strong currents (up to nAs) when membrane potential was clamped tens of mV from rest. Then, for each cell, we adjusted the voltage until light evoked no current. For CC recordings (Supplementary Figure 2) we injected current that shifted the down-state V_m to the reversal potential of inhibition.

7. The examples shown in fig1, suppl fig1 and suppl fig2 show an almost complete suppression of activation of one of the two cells. Much more dramatic than the 33% indicated in the results. This greatly reduces the significance of the results since by definition, the reduction to almost zero of the activity of one of the two cells leads to reduction in the crosscorrelogram.

By definition, cross-correlation of two signals equal in length is computed following normalization by the product of their variance and length (i.e., by $\frac{1}{n} \sigma_x \sigma_y$ where n is the length of each signal). Hence, the cross-correlation of two signals that are identical in shape but differ in their amplitude results with 1 at zero lag. Clearly noise may affect the magnitude of the correlations, but in our measurements we subtracted the shuffled cross-correlations. Hence, we have full confidence in our measurements.

Regarding the numbers, the 33% reduction is the average value for the population and not for the example pair shown in Figures 1 and SF 1 (now figure 3b and SF 2, the latter is the same pair under CC). The large distribution of the Q ratio of ongoing

activity under cortical silencing (supplementary figure 5) and the clustering of the correlation of the thalamic inputs near zero (Figure 3c) indicate that the reduction in cortical synchrony is not explained by the amount of reduction in cortical activity during LED ON condition.

Nevertheless, to illustrate that the reduction in ongoing correlation is not due to almost complete disappearance of ongoing activity in one of the two recorded cells, we have changed the scale of the figure and added here an additional example (see rebuttal Figure 1, which will be presented as a Supplementary Fig. 3).

Rebuttal Figure 1: Example traces of ongoing excitatory currents in two simultaneously recorded L4 cells during LED OFF and LED ON conditions

8. The recordings in Fig 1 show clearly recurring spindles and therefore the activity remaining after cortical inhibition should be rhythmic thalamic bursting, which is not at all apparent in the recordings. Please explain in results.

We agree that there is oscillatory activity in Figure 1. However, we have not simultaneously recorded in the thalamus to know that these are thalamic spindles. Therefore, we do not necessarily expect to see them under cortical silencing. Furthermore, even if this activity originated from the thalamus our results show that these inputs do not impinge directly on the cells that we recorded from and therefore require an active cortex.

9. Please explain in results the points with values above 1 in fig. 1i.

In these paired recordings some cells show larger activity following cortical silencing. We have no clear explanation for this result.

10. What is the point of the L4-L5 pairs? Please explain rationale in the results.

As mentioned briefly in the manuscript, studies show that L5 cells receive also direct thalamic inputs (Wimmer et al., *Cereb. Cortex*, 2010; Constantinople and Bruno, *Science*, 2013). We are the first to directly estimate the contribution of thalamic inputs to sensory evoked excitation in L5 (~19 %, figure 1d,e). Since L5 cells exhibit direct thalamic inputs and they exhibit a prominent synchronized synaptic activity with L4 cells during ongoing activity we included these pairs in our study. We clarify this point better in the text.

11. How many L4 cells did not receive direct thalamic input?

When examining the contribution of thalamic inputs as function of the recording depth for all cells, including single cell recordings, we found only 3 cells in L4 for which thalamic contribution was lower than 10% (see Fig. 1e).

12. Please show the distribution of latencies that demonstrate that every single neuron in the database could be classified as receiving or not monosynaptic thalamic input.

Please see Rebuttal figure 2 for a distribution of latencies.

Rebuttal Figure 2: latency distribution of L4 cells. Median is marked by a dashed black line.

13. The strong statements in the text about certainty of connections based only on latency require equally strong demonstration that latency alone is enough, otherwise a substantial amount of rewriting and reinterpretation is necessary.

Our certainty regarding the classification of the cells is based on three independent measurements: 1. The depth of the recording. 2. The latency of synaptic response to

whisker stimulation. 3. Remaining whisker evoked synaptic response during LED ON condition. We are therefore confident that no change is required in these statements.

14. On top of page 6 the authors conclude that the low TTC of L4 cells must be due to cortical synapses, however removing the cortical input with blue light does not significantly change TTC, thus the conclusion is not sustained by the data.

Although on average the TTC during LED ON is not significantly different than that measured during LED OFF, the connecting lines in this figure cross each other. Therefore, the TTC of thalamic inputs cannot **predict** those during intact cortical activity. This conclusion was stated in page 6: "In summary, we conclude that despite the prominent contribution of thalamic inputs to the response of L4 cells, the TTC of sensory response is not determined by these inputs, indicating that it must be determined by cortical synapses." In the new version we measured the Pearson correlation coefficient between the two conditions across the populations and made these statements (page 7 of the new submission):

"Notably, as the lines connecting the two conditions crossed each other, the TTC_{EE} during LED OFF condition could not be predicted from the one measured during cortical silencing. That is the distributions across the conditions were not correlated with each other ($r^2=0.07$, $p=0.557$ and $r^2=0.16$, $p=0.596$ for L4-L4 and L4-L5 pairs, respectively)."

Thus, the lack of correlation between the conditions is a strong indication that the TCC of the intact cortex cannot be predicted from the TTC of thalamic inputs.

15. About figure 2: Why do the currents in the upper cells in the three examples of figure 2 appear considerably smaller than the lower cell?

For illustration purposes only we arranged the cells in the Figure such that the lower cell in each pair exhibits larger response. A note was added in the main body of the manuscript order to clarify this issue.

16. It is surprising that the points of pair 13 are above the diagonal line, by looking at the currents it seems they should be below. Please show details of the currents that agree with the distribution of the points at least for that pair.

We thank the reviewer for this presentation error. This problem was corrected.

17. Looking at figure 3C, it seems that a single point (with negative correlation) makes the L4-L4 difference not significant. It seems obvious that the authors need more data to reach a conclusion about the role of cortical input on TTC.

As stated above the dataset was expanded to include more paired EPSCs-LFP recordings in awake animals that strongly support our conclusions that were obtained with anesthetized mice.

18. Even though variations in shunting do not correlate with the contribution of thalamic input it, shunting obviously (by definition) contributes to the value of it, and many of the values are high. The contribution numbers should therefore take shunting into account.

Although shunting may exist, its effect is not obvious (Figure 1g). For small cells that can be modeled as point neurons, activation of GAD cells, when the membrane potential is clamp at the reversal potential of inhibition, should have only a negligible effect on excitatory current. This is expected theoretically (as we also verified in VC simulations that were performed before we have started this project). At voltage clamp mode, when the cell is clamped at the reversal potential of inhibition, only poor

space clamp could result in shunting. Accordingly, under poor space clamp conditions, a larger attenuation in synaptic excitatory current should be correlated with larger reduction in input resistance. However, we observed the opposite (Figure 1g, with examples). Hence, we can safely conclude that shunting effect did not contribute to the observed reduction in synaptic currents.

Reviewer #3:

The authors try to determine the contribution of thalamic and intracortical inputs in generation of cortical synchronization by optogenetically silencing cortical firing and simultaneously recording from nearby L4 cells in the barrel cortex. However, the major point of study is not well established by the present data. A couple of essential components need to be further investigated.

1. A major missing part of the study is whether thalamic neurons exhibit correlated activity (evoked or spontaneous), with or without cortical silencing. As thalamic responses are regulated by the cortical feedback, the silencing of the cortex may directly alter the activity pattern in the thalamus. Another possibility is the cortical silencing may acutely disrupt the activity balance in the subcortical circuitry, which may be reflected by the asynchronous activity.

We thank the reviewer for his/her comment. We added a new set of data showing that the firing response to whisker stimulation of thalamic cells was not affected at all when the cortex was silenced by light. Although we did not record simultaneously from different thalamic cells, it is very unlikely that the precise spiking response have

not changed and yet correlated activity of thalamic cells was altered due to cortical silencing.

2. Another concern is on the functional meaning of cortical synchronization in anaesthetized brain circuits. Activity correlation reflects the properties of the neural network, and it would be highly subject to the modulation of the anesthesia. Would the conclusions (synchronization and the thalamic contribution) qualitatively hold for brain circuits in awake animals?

Indeed this is a very interesting question. As stated above for reviewer 1 and 2, we have recorded additional cells combined with LFP in awake animals and have arrived to the same conclusions. Namely, correlation of cortical activity in layer 4 is not determined by correlation of their thalamic inputs.

Dear Dr. Ranade,

I have received the decision letter from Dr. Arguello where he suggested to transfer our paper from Nature Neuroscience to Nature Communications. In our study, titled "*Local and thalamic origins of ongoing and sensory evoked cortical correlations*" we used in-vivo whole cell recordings and optogenetic in order to reveal the role of thalamic versus cortical inputs in the emergence of synchronous activity in the barrel cortex.

The origin of synchronized ongoing and evoked activities in primary sensory cortical areas is under dispute. In one view, the variability in sensory responses mostly depends on subcortical processing of sensory inputs, whereas in the second view synchrony emerges locally in the cortex. We addressed this question by performing simultaneous whole cell patch recordings from pairs of cells in layer 4, which receive powerful inputs from thalamic cells as well as local cortical inputs. We isolated the thalamic inputs of these pairs by silencing cortical firing using optogenetics and compared the synchrony of these inputs to those measured when the cortex was intact.

Our results show that in contrast to the synchronized activity of synaptic inputs in the intact cortex, thalamic inputs when the cortex is silenced are asynchronous during ongoing activity and they exhibit lower trial to trial correlation ('noise correlation') in most of the pairs that we recorded. We suggest, therefore, that synchronized local cortical activity emerges locally in the cortex. These results, obtained in anesthetized mice were further supported in awake mice by performing patch recordings in individual cortical cells simultaneously with nearby LFP recordings.

We submitted our paper to Nature Neuroscience as a Brief Communication and converted it to a full length article after a substantial revision. In this revision we addressed all the original concerns of the reviewers, including adding data from awake mice. However, the paper was rejected and we can only suspect that the decision was based on comments made by one of the reviewers ('reviewer # 1'), which raised concerns regarding the implications of synchronous responses in processing information and computations in the barrel cortex. However, since these comments are far from being specific enough to be addressed experimentally, we have modified the discussion of our paper in order to emphasize the significance of our findings.

We note, that in the rejection letter from Dr. Arguello, it was mistakenly written that our paper is a brief communication, although it was resubmitted and reviewed as a full article. Our point by point response to the recent reviews appears below. Please note that the paper was uploaded to BioRxiv ('<http://biorxiv.org/content/early/2016/06/13/058727>') and several small modifications were made based on comments obtained from this submission.

Yours,

Ilan Lampl

Point by point

Reviewer #1:

Remarks to the Author:

The authors did more experiments including in awake animals to support their main conclusions which strengthened quite significantly their manuscript. The data are now more robust demonstrating their findings. However, my main concern is still that the significance of these findings in terms of information processing are not clearly flashed out for the general reader of NN. How does, the emergence of synchronous responses attributed to cortical processing help us understanding how information is processed in that part of the brain and how does this work advance our understanding of the computations of the barrel cortex?

We thank the reviewer for acknowledging that we have strengthened our manuscript. How cortical synchronous activity contributes to information processing in the cortex is indeed a very important question. We believe that our findings, showing that local circuits are involved in shaping the variability and synchrony in this part of the brain, are crucial for our understanding of cortical computations. We therefore added a paragraph in the discussion regarding this important comment.

“Indeed, the amount of possible information that could be extracted from a neuronal population is constrained by the noise correlations of the response⁵⁷. Therefore the emergence of synchronous activity by local cortical mechanisms and the impact of these circuits in determining noise correlations is essential for understanding how information is processed in the cortex. “

A reference (57) was added: Kohn, A., Coen-Cagli, R., Kanitscheider, I. & Pouget, A. Correlations and Neuronal Population Information. *Annu. Rev. Neurosci.* (2016). doi:10.1146/annurev-neuro-070815-013851

Reviewer #2:

I am impressed by the thorough revision of the manuscript with inclusion of awake data and by the thorough response to my comments. I find this version of the manuscript an important addition to our understanding of thalamocortical circuits.

We greatly appreciate the positive comments regarding the revision and importance of our manuscript.

For fairness, on page 5, where it says, "during ongoing activity" it should say: " during ongoing activity under anesthesia".

We agree with the reviewer, and have changed the sentence as suggested.

In addition, I think the authors should consider adding the figures of the rebuttal to the supplementary material, as they were important in forming my opinion, I presume they will be important as well for readers of the manuscript.

We again agree with the reviewer's comment and have added the above mentioned figures as supplementary figures.